# Review on the Use of Heavy Metal Deposits from Water Treatment Waste towards Catalytic Chemical Syntheses

**DOI:** 10.3390/ijms222413383

**Published:** 2021-12-13

**Authors:** Tushar Kanti Das, Albert Poater

**Affiliations:** 1Rubber Technology Centre, Indian Institute of Technology Kharagpur, Kharagpur 721302, India; tuserkantidas123@gmail.com; 2Institute of Computational Chemistry and Catalysis, Department of Chemistry, University of Girona, c/Maria Aurèlia Capmany 69, 17003 Girona, Spain

**Keywords:** heavy metals, wastewater, reutilization, catalysis, mechanism, immobilization

## Abstract

The toxicity and persistence of heavy metals has become a serious problem for humans. These heavy metals accumulate mainly in wastewater from various industries’ discharged effluents. The recent trends in research are now focused not only on the removal efficiency of toxic metal particles, but also on their effective reuse as catalysts. This review discusses the types of heavy metals obtained from wastewater and their recovery through commonly practiced physico-chemical pathways. In addition, it covers the advantages of the new system for capturing heavy metals from wastewater, as compared to older conventional technologies. The discussion also includes the various structural aspects of trapping systems and their hypothesized mechanistic approaches to immobilization and further rejuvenation of catalysts. Finally, it concludes with the challenges and future prospects of this research to help protect the ecosystem.

## 1. Introduction

The dizzying advances of industry and technology have become one of the serious problems in our environment, as they can directly or indirectly leave a big impression on human health and other life systems [1]. Research based on the removal of heavy metals containing industrial effluents from the freshwater bodies has gained more attention due to the toxic effect on the life cycle of living beings [2]. But first, what are, by definition, heavy metals? Simply put, they are generally considered to be those metals with relatively high density, molecular weight, or atomic number. They encompass up to 96 out of the 118 known chemical elements; in detail, only the alkali and alkaline earths, together with aluminum, are excluded from the overall metals list due to their relative low density. Heavy metals have long term effects on living beings, and sometimes lead to death. Other types of pollutants present in wastewater, such as plastics, plant nutrients, pathogens, and synthetic organic & inorganic chemicals, are not as harmful to the environment like toxic heavy metals [3]. This distinguishes the characteristic property of heavy metals, namely environmental enrichment effects and challenging biodegradation, completely isolating them from other pollutants [4]. Beside these, if heavy metals, even with a very low dosage, infiltrate living beings through the ecosystem, they can cause harmful effects due to their high stability, solubility, and their easy migration into the aqueous medium [5]. Essentially, heavy metals can be easily absorbed on the surface of the microorganism, and sometimes break down the cell wall to penetrate inside the cell. When metals enter a microorganism, they can be transformed into poisonous materials by the chemical reaction caused by the digestive mechanism of microorganisms. Following this pathway, these metals can easily disappear into the ecosystem [6]. On the other hand, these metals also have a commercial value as a substantial non-refillable source [7]. The main principle of waste depreciation is the reuse of waste materials as a catalyst and the catalytic reaction carried out under normal atmospheric conditions (i.e., at room temperature and normal atmospheric pressure) [8]. Regarding the environment, the use of metal waste as a catalyst and the reaction in aqueous medium is highly charming [9,10]. Hence, the primary concern is not only to effectively regulate heavy metal pollution, but also requires the reuse of waste materials as a potential resource for catalytic applications.

Heavy metals are defined as the metals having an atomic density greater than 4 ± 1 gm/cm^3^ [11]. Here, it should be pointed out that metals in themselves are not toxic agents, but, depending on how they are combined, and especially in water, they present a major problem, and are almost always toxic in this environment. These non-degradable metals are the most dangerous contaminant found in industrial wastewater, and can easily change the physico-chemical and biological quality of freshwater bodies [12]. In 1987, the Environmental Protection Agency (EPA) of the United States of America announced a list of toxic elements found in wastewater, and it was found that most of the heavy metals mentioned are present on the list [13]. Depending on the properties of metals, they are classified into four categories [14]:Toxic heavy metals (e.g., −Hg, Cr. Pb, As, Sr, Si, Ag, and Ti);Radioactive metals (e.g., Tc, U, Rn, Th, Ra, and Am);Metals essential for metabolism (e.g., Mo, K, Ca, Fe, Ni, Cu, and Zn);Metals detection of biologic effectiveness (e.g., Ge, Sb, Te, Po, and B).

Currently, the recent research trend is more focused on the removal of type 1 & 2 metals and their reuse for various fruitful purposes. The primary source of these metals are various industries, such as coating, metallurgy, paper, mining, tanning, agricultural chemicals, battery manufacturing, and other industries [15]. According to several survey reports, developing countries are mostly affected by heavy metal pollution [16,17,18,19]. There are various regulatory bodies, such as the World Health Organization (WHO), EPA, and the Food and Agriculture Organization (FAO), which set concentration limits of these heavy metals in water; above this critical limit, the concentration is considered hazardous for living beings [20,21]. Thus, understanding the influence of heavy metals on environment is highly essential for developing a specific method to remove these metals from wastewater, and further make use of them as valuable resources. On the other hand, there is a certain fever or tendency to move processes in the industry, and also in basic research [22], including predictive catalysis [23,24,25], that are carried out with transition metals of the second and third row by others of the first, due to being more abundant [26], but in most cases with less efficiency [27,28,29,30]. Erroneously, this is also associated with lower toxicity, when there really is no clear connection to make such a claim. Another point to clarify is that it is metals in themselves are not toxic, but their toxicity is dependent on the context in which they are found. Thus, while existing in soil or living organisms, these heavy metals are essential, and therefore far from any toxicity (the third point of the previous list), but everything is reversed when they are in wastewater [31].

Many efforts have already been made to eliminate the environmental hazard of wastewater, and to reuse heavy metal nanoparticles as a catalyst for various applications purposes [32,33,34,35]. However, the diversity of heavy metals contaminates wastewater, and its complexity in the existing structure leads to the treatment of wastewater being a highly challenging task. Recently, various emerging methods of wastewater treatment have been considered given the above-mentioned facts, which can recognize both the easy recovery of heavy metals, and those used for different purposes of catalytic applications [36,37,38,39]. The progress of this type of research would shed new light on the treatment of industrial wastewater polluting heavy metals.

In this review, one could wonder why we omitted that one of the main pollutants are the metallurgical refineries that produce heavy metals [40], since they generate tons of waste. Take for instance, before the Second World War, the production of materials produced an exaggerated rate of waste. Take, for instance, that for 1 ton of steel, 1.4 tons of waste were generated; and in the case of aluminum, 15 tons of bauxite and 3 tons of red mud. In addition, nature is polluted both by the refineries themselves and by landfills, or sludge deposits of these refineries. However, this is out of the scope of the current work that aims to functionalize the waste through its treatment In detail, the source, toxicity limits, treatment methods, and reusability techniques of heavy metals polluting wastewater are compiled in an orderly manner, with an understanding of the properties of heavy metals, which illustrate the status current treatment methods and future prospects in this research area.

## 2. Source of Heavy Metals

There are various sources of heavy metals in our environment such as (1) natural resources, (2) mining, (3) electronic waste, (4) power plants (5) agricultural sources, (6) industrial sources, (7) household waste, and (8) other resources. The sources of heavy metals are shown schematically in Figure 1. The separation of the source and treatment of heavy metal pollution in many of the categories discussed below may be the subject of debate, and surely some categories could be combined, such as the remains of manufacturing companies and of mines, within a wider industrial section. Therefore, we do want to emphasize that we have tried to deal with each of the problems in a particular way, even though they are part of a larger whole.

### 2.1. Natural Resources

The natural geographical phenomena, such as volcanic eruptions, the rupture of rocks & minerals release enormous heavy metals into rivers, lakes, and oceans. Rocks and minerals are the primary resource of heavy metals in the environment and, depending on the types of minerals, there are various types of heavy metals which exist as trace elements. The process by which heavy metals are incorporated into rocks is called isomorphic substitution [41,42].

The uncontrolled smelting of ores or mines in open atmosphere also releases a small number of heavy metals into the environment. With the progress of industries, natural resources are used as source of metals; during the extraction of these metals, heavy metals are accumulated in water bodies. The discharge of various agricultural and household wastes can lead to the massive accumulation of heavy metals.

### 2.2. Mining Operations

Mining operations can pollute the ecological system with various heavy metals through a range of mechanistic pathways, such as the physical eruption of land, splashing of mines, and acid waste streams [43]. Due to the government’s lack of awareness, mining activity operated on small scales is taking place around the world, and the operation has become widespread in developing countries [44]. Most small-scale mining operations are illegal, and can cause severe environmental pollution due to poor practice by the workers during the processing of minerals. It has already been reported the environment is mostly polluted by the heavy metals due to a lack of regulation on small-scale mining activity [45]. In particular, the main heavy metals found in the mining areas are Fe, Mn, Cu, Zn, Pd, Co, As, Ni, and Cd, whereas the study of the literature revealed that Fe and Mn were present at high levels in the soils [46].

### 2.3. Electronic Waste

With the rapid growth of information technologies and the short life of the electronic products, the waste of electronic materials has become one of the expanding problems of solid waste [47]. Toxic heavy metals such as Pd, Cu, Sn, Ag, and Zn are mostly found in electronic waste of wire, cables, printed circuit boards, plugs, and chips among other items. Thus, the recycling of electronic waste is very demanding in developing countries and, during the uncontrolled recycling process, exposes these toxic heavy metals to the environment [48]. The highly informal way of burning electronic waste outdoors in front of workshops has polluted more than other forms. To better understand the heavy metals pollution produced by electronic waste, research is needed on the surface distribution of these metals. Few studies already reported that pollutants remain in the same place, and do not migrate to other places [49,50]. The risk of heavy metals pollution is higher when the electronic industry uses metals as its major elements. The growth of electronic waste increases at the rate of 2 million tons (Mt)/year, with the incremental use of electronic items. As per a survey report, E-waste reached 53.6 Mt in 2019, and Asia (24.9 Mt) is the largest continental contributor, with Europe (12.0 Mt) being the smallest contributor to electronic-waste. As a result, most countries have imposed strict guidelines on to the producers, end-users, and recycling techniques [51,52]. This increase in volume of E-waste might be a serious threat for social life, as these materials consist of toxic heavy metals which negatively impact human health. The adverse effects of E-waste on the human health are dermal disorders, cancer, cardiovascular problems, respiratory effects, and neurological disorders [53,54]. Consequently, it is urgent that the necessary steps on the effective management on the recycling of e-waste through the proper system to achieve the sustainability of the ecosystem are taken.

### 2.4. Power Plants

The combustion of coal in thermal power plants is another source of environmental pollutants by heavy metals due to the generation of a large amount of ash that can contaminate nearby water bodies where the ashes are deposited. Toxic heavy metals present in the ashes include Mo, As, Cr, Mn, Cu, Ni, Co, Pb, and Zn [55,56]. On the other hand, the condensed flue gases emitted by the boilers of heavy metal (Hg) power plants can infect the water bodies through air discharge. The risk of metal pollution is much higher when thermal power plants use coal to generate energy [57]. For example, Sushil et al., analyzed the heavy metal in coal fly ash and coal bottom ash on three different places of power plants. They reported that most of the heavy metals were present in the coal fly ash rather than in the bottom fly ash. They found higher concentrations of Cr, Mn, Pb, Zn, Cu, and Ni as heavy metals in the coal fly ash, as well as in the coal bottom ash [58]. The energy industries, such as nuclear and mining industries, create huge amounts of toxic radionuclides and heavy metals. The nuclear power plants have discharge large amount of Cu, Cr, and Zn as heavy metals. Nuclear power plants consume a substantial amount of water during their operation, and after their operation, heavy metals are discharged with this consumed water, which finally pollutes the soil, as well as aquatic systems [59].

### 2.5. Agriculture Sources

To improve the growth of plants, various inorganic and organic fertilizers are added to agricultural soils that use heavy toxic metals as the main ingredients [31]. These fertilizers can either expose these toxic metals to the soils, or run off with water during rainy seasons to contaminate water bodies. Fertilizers contain Cd, Cr, Zn, Pb, Mn, Cu, and Ni as heavy metals, and, depending on the source, the amount of these toxic metals varies [60]. On the other hand, agricultural soils are also enriched with Mn, Zn, Cu, and Co with animal manure. The concentration of heavy metals in the agricultural soils is based on the rate of involvement of the donor substrate and soil type [61]. Several pesticides (consisting of heavy metals such as Pd, As, Cd, Mn, and Hg) are also applied to the soil to kill the pulp, which is a minor contamination of the soil [62].

### 2.6. Industrial Sources

Industrial sources, including the chemical, alloy, paint, glass, pulp & paper mills, textile drying & printing, oil refining, and leather tanning industries are other major sources of heavy metals. These toxic metals are primarily dispersed in the ecosystem by industrial effluents that emerge from the industry combined with wastewater. The heavy metals discharged from these industries are As, Cd, Cu, Cr, Pd, Hg, Ag, Ni, and Zn [63]. It has already been investigated that a higher concentration of heavy metals is found near industrial areas, and contamination was determined by the geochemical index [64].

### 2.7. Domestic Waste

Household waste is basically discharged into the nearest sewage, and this waste gradually flows with water and finally combines with water bodies to pollute the environment. For example, detergents used for washing contain heavy toxic metals such as Mn, Cr, Co, Sr, and Zn, and if water used for washing has not been properly treated, it can pollute the environment. The problem is more serious in urban areas due to ignorance [65].

### 2.8. Other Sources

Two main sources of heavy metals generated during human activities are the burning of coal in an open atmosphere, and the removal of corrosion from waste products that discharge Cr, Cd, Hg, Mn, Ni, Cu, Pb, and Zn into the environment [66]. The burning of oils and wastage tires in front of workshops emits heavy metals (e.g., Pb, Ni, Zn, and Fe) into the soil [67]. Gasoline, which is enriched with Pb as one of its burning elements, releases these toxic metals into the atmosphere. On the other hand, studies have revealed that heavy metal pollution from municipal waste garbage is increasing daily, and developing countries are mostly affected by this type of pollution [68].

## 3. Concentration of Heavy Metals

### 3.1. Heavy Metal Concentration in Wastewater

Heavy metal-based compounds are highly soluble in aqueous media, and these are easily absorbed by living bodies. Heavy metals invade the food chain system through this absorption process, and finally the process ends with the accumulation of toxic heavy metals in human bodies. With the exception of Pb, the chance of heavy metals infiltrating wastewater is almost 100 wt.%. The order of concentration of heavy metals in the wastewater in 2003 was as follows: Fe > Zn > Cu > Ni > Mn > Pb > Cr > Cd [69]. The presence of heavy metals is clearly observed when comparing treated wastewater and wastewater discharges as industrial effluents. It has already been revealed that sedimentation and the biological process greatly affect the distribution of heavy metals such as Zn, Pb, Cr Cu, Cd, and Ni in wastewater. The distribution of heavy metal in wastewater depends on the specific metals and their chemical form, solubility, and pH of wastewater [70]. Experimental studies explored that the carbonate salts of Ni and Zn, the sulfide salt of Cr and Cu, the oxide form of Fe and Mn, the hydroxide form of Pb, Zn and Ni, and the crystalline phases of Fe and Pb are the major carriers of various heavy metals [71]. The partition coefficient (K_d_) is an important parameter for determination of heavy metal in wastewater, which is defined as the ratio of suspended solid to the dissolved solid in the aqueous phase. The value of K_d_ for different heavy metals follows the order Fe > Cd > Pb > Cr > Ni > Mn. This order of value of K_d_ suggests that the highest concentration of Mn was found in wastewater, while Fe was located mainly in sludge as waste materials [72].

### 3.2. Heavy Metal Concentration in Sludge

The abundance of heavy metals in the is almost 100 wt.%, and the concentration of Zn in particular has been found in higher amounts in the sludge, whereas Cd has been found at a lower rate. The Pb concentration is higher in digested sludge due to the high atomic weight of Pb, and the degraded anaerobic digestion and degradation processes in gaseous products. It has already been explored that the concentration of Pb, Cd, Cu, As, and Zn is 50 wt.% higher in digested sludge than in undigested sludge [73]. The heavy metal content of the sludge varies in urban as well as industrial areas due to the variation of the external entrance in the different zones. Different wastewater plants have already reported that heavy metals content in sludge is eight times higher than previously reported. Among heavy metals, the Fe content is much higher than other metals [74]. For this purpose, new technologies have been adopted at wastewater treatment plants to reduce the metal content of sludge after the treatment of wastewater obtained from different sources. On the other hand, it has also been noted that the concentration of heavy metals in sewage sludge depends on the cleaning performance of each industrial plant [75].

### 3.3. Regulatory Limits of Heavy Metals

Although heavy metals are natural materials, technological activities have altered their cycles in environments. Due to this alternation in cycles, metals are consumed in different parts of the environment, which is leading to various harmful effects on living bodies. Among heavy metals, few metals are essential, such as Fe, Cu, Zn, and Mn for living bodies, but after a specific concentration, these metals have become toxic [76]. On the other hand, heavy metals such as Pd and Hg are useful for human bodies, but have a detrimental effect on human heath if they accumulate more over time. Essentially, human bodies are in contact with these metals during ingestion and breathing processes. The risk of exposure of these heavy metals to people is high, whether working or living near the industrial areas related to these metals and their corresponding compounds. In short, heavy metals are not harmful to human health until they exceed the limits of toxicity. The pathway in which heavy metals affect human bodies depends on the specific metals, but ultimately all metals produce reactive oxygen radicals, which leads to various diseases in living bodies. Thus, it is essential to establish a limit above which toxic elements were considered [77]. Based on the toxicity, the regulatory limits of heavy metals set by the various well-known organizations are summarized in Table 1.

## 4. Harmful Effect of Heavy Metals on Human Health

Heavy metal water contamination has become a major concern for human health. Consumption of heavy metals in the human body has adverse effects. Heavy metals include As, Ba, Cd, Pb, Se, Hg, Ag, Cr, and Fe, which are the main culprits for various harmful effects on human health.

Arsenic is an odorless and tasteless inorganic carcinogenic compound; exposure causes skin, liver, and lung cancers [81]. At low doses, it can reduce the production of red and white blood cells, destroy blood cells, and cause skin irritation, whereas long-term exposure of As at low levels can darken the skin, and lead to the appearance of small corn on various parts of the body. The introduction of a very high level of As can cause death [82]. Barium (Ba) is an abundant and natural material used for various industrial purposes, such as drilling mud, and the paint, brick making, ceramics, glass, and rubber industries. Short-term exposure of Ba-based compounds can cause vomiting, abdominal pain, respiratory problems, fluctuations in blood pressure, and intense muscle pain, whereas long-term exposure can alter heart rate, cause paralysis, and sometimes even lead to death [83]. Cadmium is another toxic inorganic material used in various industries such as the battery, electroplating, painting, and plastic industries. Smokers are more exposed to this metal than others, which can cause serious damage to the lungs and, sometimes, death. Short-term exposure to Cd can cause diarrhea, and long-term exposure can cause damage to the lungs, kidneys, and bones [84]. Due to various human activities, the combustion of fuels, mining, and the manufacture of lead (Pb) and its corresponding compounds can be found throughout the environment. Pb has a great nature, so its use in different industries such as paint, gasoline, pipes, and batteries has been drastically reduced over time. Long-term exposure of Pb can alter the normal function of the nervous system, cause weakness in the fingers, damage the function of the brain and kidneys, and can sometimes lead to death [85]. Selenium (Se) is a trace mineral, and is used in many industrial sectors, such as the electronic, paint, pharmaceutical, ink, rubber, and glass industries. Se is an essential trace element for living bodies, as it is necessary for cellular function. Short-term exposure to Se can cause nausea, vomiting, and diarrhea, while long-term exposure causes stomach pain and respiratory problems, including bronchitis [86]. Mercury (Hg) is used as a metal form, and in other organic and inorganic complexed forms. It is used in thermometers, dental fillings, batteries, and light bulbs. A low level of Hg exposure can damage brain and kidney function, while a short level of exposure causes nausea, vomiting, diarrhea, skin rashes, and eye irritation. In addition to these, the nervous system of the human body is highly sensitive to all forms of Hg [87]. Silver (Ag) is also used in its metallic form, as well as in its complexed form. It is employed in jewelry, electronic industries, photographic films, and antimicrobial agents. Although Ag has antimicrobial properties, exposure to high levels of silver can cause argyria, skin irritation, throat irritation, respiratory problems, and skin rashes [88]. Although Ag is absorbed and digested in the soft tissue, various studies have reported that when it infiltrates the central nervous system, it can cause neurotoxic damage. Ag primary enters the human body through the genital tract and during inhaling, where it deposits in the lungs [89]. Chromium (Cr) compounds can easily bind to soils and do not flow easily with water. Therefore, it is found mostly in the sediments of water. This metal is used in electroplating, cement, papers, wood preservation, paint, and rubber industries. Cr with (+III) oxidation state is an essential element, whereas with (+VI) oxidation state, it is toxic in nature. Cr can cause skin rashes, allergic reactions, damage to the function of the kidney and liver, and respiratory problems [90]. On the other hand, iron is one of the essential elements for the human body, and its deficiency can cause anemia. It has already been explored that a high dose of Fe could be the root of several diseases [91]. Moreover, Mn can affect the respiratory system, as well as the brain. Mn can cause Parkinson’s disease, pulmonary embolism, and bronchitis, and long-term exposure of Mn can lead to impotence [92]. This discussion has clearly mentioned that all the heavy metals are toxic by nature above a certain concentration. Therefore, it is highly important to remove materials from environments and use it as energy sources for other applications.

## 5. Recovery of Heavy Metals from Wastewater

There are lots of conventional methods available to removal heavy metals from wastewater. They are classified into four categories: electrochemical treatments (such as electrocoagulation, electro-flotation, and electro deposition), physico-chemical processes (such as chemical precipitation, and ion exchange), absorption processes (such as activated carbon, carbon nanotubes, and wood sawdust absorbents), or current developed methods (such as membrane filtration processes, photocatalysis processes, and nanotechnology) [93,94]. The schematic diagram of trees in this conventional heavy metal removal process is presented in Figure 1.

### 5.1. Electrochemical Treatment

Electrochemical wastewater treatment for the removal of heavy metals is one of the most efficient and compact methods. However, from an economical perspective, the method is very expensive due requiring large amounts of electricity. For the treatment of some industrial effluents, such as pollutants in a refectory, this process is essential [95]. Electrochemical treatments are basically classified into three categories: (1) electrocoagulation, (2) electrodeposition and (3) electroflotation. The direct comparison between methods is difficult, due to the lack of numerical values, but some of the main disadvantages of each are announced below.

#### 5.1.1. Electrocoagulation

Electrocoagulation is a very simple and productive method for wastewater treatment. However, the method is not considered as adequate due to poor reactor design and problems on the part of electrodes. For this reason, the method is very useful for treating small batches of wastewater with improved technology. This process was first employed in 1909 using aluminum and iron electrode in the United States of America (USA) [96]. In this process, the reactor is an electrolytic cell in which one is a cathode, and the other is an anode. Electrodes consist of either similar or different types of materials. The electrolytic reactor is immersed in the wastewater solution, and, after the application of electric field, the suspended heavy metal pollutants coagulate in the water, forming a large mass called sludge. These sludges are removed through the filtration process, and the heavy metals are recovered. The advantage of this process is that the sludge produced is less bonded to water, and much more stable than other processes [97]. It has already been explored that for the removal of various heavy metals, different materials are used as electrodes to improve their efficiency. For example, an Al electrode was used to separate Co and Mn materials from wastewater solution [98]. The process is environment friendly, as the method uses electrons to clean the environment without adding chemical additives to the wastewater solution. On the other hand, this method fails to negatively to treat acid wastewater, and, furthermore, since the sediment contains heavy metals, further processes are required, including dehydration or neutralization.

#### 5.1.2. Electrodeposition

This method is very efficient in recovering heavy metals from wastewater. The advantage of this method is that no additional chemicals are needed for the process, no sludge-like materials are produced, the process is highly selective in nature, and the dissolved heavy metal ions directly deposited on the electrode are transformed into solid materials. The process is done through one step of the oxidation and reduction method, in which heavy metal ions are reduced to deposited on the surface of cathode. The reactor of this process consists of an electrolytic cell made of a cathode and an anode and an electrical field is applied to conduct the process of oxidation and reduction simultaneously [99]. The efficiency of the process depends on the concentration of heavy metals in the wastewater, the temperature, pH of the solution, and the complexed forms of these metals. The process can be applied to non-aqueous solutions, and performs better than the aqueous solution [100]. Wulan et al., employed two types of batch reactors. One is a partitioned reactor, and another one is a single chamber reactor for the electrodeposition process. They found that the partitioned reactor performed 26.89% better than the single chamber reactor toward the electrodeposition process [101]. Take, for instance, oxidation, through steam stripping, air stripping, or activated carbon absorption, can lead to the removal of any toxic byproducts created by chemical oxidation. The oxidation treatment is always used for the pretreatment of heavy metal wastewater containing organic compounds. Figure 2 demonstrates a schematic mechanism of advanced oxidation processes (AOPs) in wastewater treatment, consisting of the generation of hydroxyl radicals [102,103].

#### 5.1.3. Electroflotation

This process is utilized to treat wastewater that contains a very low concentration (below 50 mg.m^−3^) of heavy metals as pollutants. The process was first reported by Elmore in 1904, and, since then, the process has been widely applied due to the ease of design and operation of the reactor, its low cost of operation, and its easy installation due to the small size of the instrument [104]. The separation process of electroflotation occurs in three steps: in the first, metal pollutants are captivated towards the electrodes, and the oxygen and hydrogen formed during the electrolysis will act as the surface for the absorbent of heavy metals that ultimately creates flakes. In the second step, the separation of the settled flakes will take place. Finally, the heavy metals of the settled flakes will be recovered through filtration [105]. The efficiency of this process is influenced by various parameters, such as the size of the bubbles during electrolysis, the design of the cell, the type of materials used for the manufacture of electrode, the current density, and the temperature and pH of the solution [106]. Although the process has many advantages, there are already many limitations of this process. Therefore, to overcome the process of limiting electroflotation, it is sometimes combined with the process of electrocoagulation and electrodeposition, which can improve its efficiency to separate pollutants [107].

### 5.2. Physico-Chemical Process

The physico-chemical process of wastewater treatment is classified into the following two categories: (a) chemical precipitation, and (b) ion exchange.

#### 5.2.1. Chemical Precipitation

This method is the most widely used for wastewater treatment, as the process is very simple and easy to install anywhere. This method requires a large number of chemicals to reduce the metal ion in the form of precipitation. As the process uses a large number of chemicals for reduction, the chemicals used can become yet another source of pollutants [108]. In this process, the chemicals react with heavy metal ions to transform them into insoluble solid materials, and then, the solid phases are separated from the liquid phases either by filtration or sedimentation. pH readjustment is a crucial factor, as the basic medium is always preferred for precipitating heavy metals from wastewater solution, depending on the heavy metal compound that may form. The metal ions present in the aqueous system are either reduced or oxidized to their corresponding valence state [16]. Heavy metals precipitate mainly in the form of hydroxide or sulfur. In the process of hydroxide precipitation, various coagulation agents, such as iron salts, alumina, and polymers, are used to precipitate heavy metals from wastewater solutions. In addition to these alkaline reagents, they are also employed to precipitate heavy metals, as the basic pH decreases the solubility of heavy metal ions in aqueous solutions. After precipitation, the forms of heavy metal hydroxide are separated from the wastewater through simple filtration methods [109]. On the other hand, the idea of the sulfide precipitation process is almost similar to that of the hydroxide precipitation process, where sulfides are applied to precipitate heavy metals from wastewater solutions. However, the main problem with sulfide precipitation techniques is the toxicity of sulfides and hydrogen sulfides (H_2_S) gases that occurs during the precipitation process [110]. Although the process is simple, the separation process of the next stage, the slow precipitation of the metal ions and the increased cost of the filtration process, cause the techniques to be unrealistic [111]. In addition, this method fails to treat wastewater with high acid content, since it produces a large quantity of toxic sludge that needs to be treated with chemical stabilization.

#### 5.2.2. Ion Exchange

The process is defined as a reversible ion exchange between solid and liquid phases. The process begins with the ion interchange reaction of similar size, followed by the physico-chemical absorption of heavy metal ions by the various functional groups present in the ion exchange columns. Finally, hydration takes place on the surface of the pores of the absorbent [112]. The process depends on various parameters such as pH, temperature, ions, and the concentration time of the absorbent and sorbent [113]. In this process, resin is used that absorbs similar types of heavy metal ions from the wastewater solution through the ion exchange process. Subsequently, the separation of the resin from the wastewater solution is recovered from the heavy metals of the resin by elution using suitable chemical reagents [114]. Resins with a functional group, such as the sulfonic acid group, are able to capture the heavy metal ion using ion exchange techniques and the process take place as follows:*n* RSO_3_^−^ H^+^ + M^n+^ = *n* RSO_3_^−^ M^n+^ + *n* H^+^(1)
where RSO_3_^−^ H^+^ and M^n+^ denote resin substance and metals ions present in the solution. RSO_3_^−^ M^n+^ designates metal ions absorbed resin, and *n* denotes the number of metal ion oxidation state and the number of functional groups attached to the polymeric resin [115].

This process is very specific in nature, and is used for the efficient removal of heavy metals from wastewater solutions. It has already been studied that the more acidic functional groups are more efficient in removing heavy metals from wastewater solutions. The advantages of the ion exchange process are that it has a low installation cost, a low sludge production, a highly selective process, and an ease of recovery of precious metals [116]. Among all of the suitable materials, synthetic polymeric resin such as styrene-divinylbenzene and Ambersep 132 basic resin, polymeric gels, gel-like cation exchange resin, and sponge-like macrospores resin are used for the recovery of heavy metals from wastewater [93,117]. However, this method also has the drawback that, at high acid levels and high concentrations of heavy metals, a pretreatment is required.

### 5.3. Absorption Process

Absorption is also a well-known method used to reduce the concentration of metals in wastewater [118,119]. The basic principle of this method is that the mass is exchanged between the liquid and the solid phases. The absorption of heavy metals from the wastewater solution to the absorbent is performed through three stages: (1) diffusion of heavy metal ions from the bulk of wastewater solution to the surface of the absorption; (2) absorption of heavy metals with other pollutants on the surface of the absorbent; and (3) infiltration of heavy metals into the absorbent structure [120]. The absorbent should have a large surface area for absorbed heavy metals on its surface and, as the absorption process is reversible in nature, the absorbed heavy metal can be regenerated by the desorption process from the absorbents [121,122]. The most widely used absorbents for the absorption of heavy metals are activated carbon (AC), carbon nanotubes (CNTs), and wood sawdust, which are discussed below. On the other hand, zeolites that are naturally occurring crystalline, hydrated aluminosilicates of alkali and alkaline earth cations, are useful amendments to bind with heavy metals [123]. Natural and synthetic zeolites are used to remediate the soil from heavy-metal contamination. There are five different types of synthetic zeolites that bind with Cd and Zn, including mordenite-type, faujasite-type, zeolite-X, zeolite-P, and zeolites-A. Among 40 natural zeolites, only seven (clinoptilolite, chabazite, mordenite, erionite, ferrierite, analcime, and philipsite) have been exploited so far [124].

#### 5.3.1. Activated Carbon (AC)

The manufacture of activated carbon (AC) is realized in two stages: the first step is carbonization at high temperature in an inert atmosphere, and in the next stage, it is activated [125]. The specific surface area of AC varies from 1260–3250 m^2^ gm^−1^, which is important for absorption [126,127]. AC is comprehensively used to remove toxic heavy metals from industrial effluents from wastewater. It is also reported that the absorption of ACs depends on various parameters, such as the preparation method of AC, the type of heavy metal absorbed, the absorption temperature, the pH of the aqueous solution, and the concentration of heavy metal ions in wastewater solutions [128].

#### 5.3.2. Carbon Nanotubes (CNTs)

Carbon nanotubes (CNTs) are very popular for the absorption of toxic heavy metal pollutants. However, the process by which CNTs absorb pollutants is not yet clearly understood. Research to find the mechanisms of absorption is still ongoing [129]. As a CNT material is toxic in nature, discharges into the environment increase the risks of pollution. Therefore, researchers have been trying to modify CNTs with calcium alginate that reduced the chances of CNTs discharges into the environment [130]. Li et al., revealed that the combination of AC and CNTs can increase the absorption efficiency by 10% than pure CNTs [131].

#### 5.3.3. Wood Sawdust

Wood sawdust, an economic absorbent, is obtained during the mechanical processing of wood. It consists of cellulose and lignin that have the ability to bind ions from the solutions due to the ability to bind their functional groups such as hydroxyl, carboxylic, and phenolic groups [132]. Marina et al., have already explored the efficiency of wood sawdust absorption towards various heavy metals, and revealed the following absorption efficiency order Cu > Zn > Cd [133]. Gradually, interest based on absorption on wood sawdust has been increasing due to the environmentally friendly nature of the absorption process.

### 5.4. Current Methods

#### 5.4.1. Membrane Filtration

The membrane is defined as a layer of porous or non-porous structure which comes into contact with two phases (homogeneous or heterogeneous) to separate pollutants of various sizes [134]. This process has a higher removal efficiency and lower energy consumption than other conventional methods in operation. This advantageous property provides a large space for application of this process and, among them, is the removal of heavy water from wastewater treatment. The process is used due to the ease of its separation process. Membranes applied to the separation of various heavy metals are different and, according to the separation process, the fabrication and structure are different [135]. According to the separation process, membranes are classified into three categories: liquid membrane, pressure driven, and hybrid membrane. There are different parameters, such as the materials used, the types of pores, and their sizes and composition, that can affect the effectiveness of a membrane. The materials must be selected in such a way that they have a high productivity, with excellent resistance to chemicals, as well as fewer defects in their structure. The material used to make membranes includes ceramics, polymers, metals, and hybrid composites materials [136]. Among them, polymer-based membranes have been found to be attractive due to low manufacturing costs and porous structures [137,138]. On the other hand, ceramic-based membranes have been found to be more attractive than polymeric-based ones, due to small pore size, excellent mechanical properties, and high chemical and thermal stability. However, due to their high manufacturing process cost, ceramic-based membranes are only used for special purposes [139]. Depending on the pore size, the permeation behavior and the applied pressure separation of membranes are classified into five categories: (1) reverse osmosis (RO), (2) ultrafiltration (UF), (3) microfiltration (MF), (4) nanofiltration (NF), and (5) electrodialysis (ED) [140]. Different membranes are employed to separate heavy metals from wastewater, depending on the size of toxic metal pollutants.

#### 5.4.2. Photocatalytic Process

This is a new technology to purify water from various pollutants. Research based on this technology to treat wastewater is now in full swing [141]. This technique uses semiconductor material with a specific wavelength of light for the treatment of pollutants present in wastewater. This has been an attractive process, as the technique did not use other chemicals to treat pollutants [142]. The main advantage of this process is that it has a simple design, an inexpensive method, and high efficiency in the removal of pollutants. In addition to heavy metal treatments, the process has been applied for many purposes, such as the disinfection of microorganisms, the degradation of organic compounds, and the manufacture of chemical fuels [143]. The materials used as photocatalysts include mostly TiO_2_, ZnO, CeO_2_, WO_2_, CdS, ZnS, and WS_2_ [144]. The photocatalytic process for the treatment of heavy metal contaminants is performed in five stages. In the first stage, the heavy metal ions gradually enter the surface of the solution, and are then absorbed on the surface of the photocatalyst. In the next stage, the photocatalytic reaction takes place in the presence of a specific wavelength of light. Later, the heavy metals are converted into the desired products, and transferred into the interfacial region of the catalyst and the solution surface [145]. When semiconductors radiate with a particular wavelength of light, the electron in the valence band (VB) will pass to the conduction band (CB), creating a hole in the conduction band. This created electron and hole are responsible for the reduction and oxidation of heavy metal ions present in the wastewater solution to transform them into desired products [146]. The photocatalytic mechanism is schematically presented in Figure 3. The deposition of heavy metals in the interfacial regions depends on the type of heavy metals. When the metal is deposited, the metals can be recovered through mechanical and thermal process. For a specific condition, the following reactivity of the heavy metals was found Ag > Pd > Au > Pt >> Rh >> Ir >> Cu = Ni = Fe [147]. Besides the advantageous properties of the photocatalyst, there are still limitations of this process, such as the recombination of holes and electron pairs, side products being produced, and predetermination in the absorption of the specific wavelength of light [148].

#### 5.4.3. Nanotechnology

The materials that have a dimension in the range of nanometer exhibit distinguished properties compared to that of their corresponding bulk counterpart. Nanomaterials, due to their high specific surface area, are extensively applied in the treatment of heavy metals present in wastewater [150,151,152]. However, during this process, the effect of discharge of nanoabsorbents on the environment should be considered. The nanotechnology-based process has overcome the shortcomings of previous technologies and, in addition, the process provides greater efficiency and a low generation of waste materials [153]. The treatment of heavy metals present in wastewater can be treated using nanomaterials that are divided into two classes: one in situ, and the other ex situ. In the in situ treatment, nanomaterial-based technology has been applied to the place where it is polluted by heavy metals, whereas in the ex situ technologies, the wastewater solution is moved to other sites for treatment [154,155]. Three types of nanomaterials are used to treat heavy metals present in wastewater, namely adsorptive, reactive, and hybrid magnetic particles. Nanomagnetic oxides are an example of the adsorptive type of nanomaterials, and their effectiveness depends on the state of the systems and the nature of heavy metals [156]. Zero valance iron nanoparticles are an example of the reactive type of nanomaterials. They react with heavy metals either to degrade it, or convert into nontoxic products [38]. The last type of material is called hybrid magnetic, as it consists of two materials, including one of a magnetic nature. They are also extensively applied in the treatment of heavy metal pollutants due to the low cost, ease of the fabrication process, and high absorption capacity, as well as removal efficiency, since being magnetic in nature aids separation after the process [157,158].

The processes discussed above are employed in the treatment of heavy metals present in industrial effluents, and depend on the type of heavy metals the process is applied.

## 6. Effect on the Characteristic Water Properties in Presence of Heavy Metals and Their Removal

It is fundamental to discuss the properties of water after contamination with different toxic heavy metals. Parameters such as pH, temperature, ionic strength, and natural organic matter are crucial for consideration when water is polluted with hazardous heavy metals. The quality of the water affected by these parameters can alter the efficiency of any treatment process. The parameters are briefly discussed in the following section according to their importance [159].

### 6.1. Effect of pH

The pH of the water plays a significant role in the presence of heavy metals in the aqueous state. The solubility of heavy metal ions in water depends on the pH of the solution. In general, at a very low pH value, the solubility of heavy metal ions in the aqueous system is very high since it exists in the cationic state, whereas with an increase in pH, i.e., above pH = 7.00, the solubility of metal ions existing in the aqueous system decreases due to the formation of hydroxide complexes. In addition, the pH of the aqueous system can affect the process by which heavy metals are recovered from wastewater solutions (as discussed in previous sections) [160].

### 6.2. Effect of Temperature

Temperature is another important parameter that must be considered when evaluating the behavior of water contaminated by heavy metals. Chen et al., revealed that the efficiency of heavy metal removal increased at higher temperatures, and also proposed temperature-based mechanisms [161]. Malkoc et al., also studied the efficiency of removing the temperature dependence of Ni(II) ion from tea waste, and they found that when the temperature increased from room temperature to 60 °C, the removal efficiency was increased by 22% due to an increase in the number of absorption sites, as well as the mobility of heavy metal ions in the solution of tea waste [162]. Later, Weng et al., reported that the absorption of heavy metal increases at higher temperatures due to the increased diffusion of heavy metals towards the absorption site through the interfacial boundary layers [163]. The reverse trend was also observed in a few studies. For example, Sari et al., explored that the removal efficiency of Cr ions from red algae was reduced by 12% with an increase in temperature, due to the tendency of the Cr ion to remain in the aqueous phase [164]. Furthermore, other researchers have also discovered that, with increasing temperature, there was a reduction in the efficiency of removal due to the reduction in the activity of absorbent materials [160,165]. Hence, during the evaluation of the removal of heavy metals from the wastewater, the characteristic properties of heavy metals along with the behavior of absorbent materials should be carefully considered to judge the overall temperature impression on the efficiency of the removal.

### 6.3. Effect of Ionic Strength

Ionic strength is also another important parameter which also affects the efficacy of removing heavy metals from the wastewater solution. Ferrez et al., reported presence of chloride ion creates an obstacle in removing the heavy metals ion from the aqueous solution due to the ability of heavy metal ions to migrate from the chloride complex [166]. Later, Villaescusa et al., also observed that the removal efficiency of Cu(II) and Ni(II) by the absorption process decreased with increasing ionic strength of the medium due to the formation of a chloride complex of low absorption affinity [167]. When the interaction between heavy metals and the absorbent surface is governed by the electrostatic interaction, ionic strength plays a significant role in the characteristic properties, as well as in the removal efficiency of heavy metals. According to the theory of surface chemistry, with increasing ionic strength the electrical double layer decreases, which, in turn, reduces the absorption efficiency of heavy metals by absorption [168]. The opposite trend was also reported by Yang et al., who revealed the removal efficiency of As(III) and Ni(II) also increased with increasing ionic strength of the medium (0.01–1.0 M Cl^−^ ions) due to the compromise of the complexion of the inner sphere [169]. Thus, the ionic strength of the medium must be taken into account during the removal of any heavy metal.

### 6.4. Effect of Natural Organic Matter

Natural organic matter consists of humic acid and fulvic acids that are obtained from decomposed plant and animal products [170]. This matter has a high tendency to react with heavy metals, and can change the mobility and toxicity of heavy metal ions. Even now, it is very difficult to understand the change in behavior of heavy metals after their interaction with natural organic matter [171]. Wang et al., reported that As can react with acids present in organic matter, which might provide an increase in As immobilization [172]. Du et al., explored the presence of organic matter in the aqueous solution by increasing the absorption efficiency of heavy metals such as Cd, Pb, and Zn [173]. On the other hand, Kumpiene et al., revealed that the presence of organic matter also reduced toxicity of some heavy metals, such as As and Cr, due to their conversion into nontoxic products [174]. Although the proper interaction of organic matter with heavy metal is not explored, parameters must be considered when heavy metals are separated from wastewater obtained from different sources.

## 7. Recent Progress in Reutilization of Recovered Heavy Metals from Wastewater in Catalytic Chemical Syntheses

It is now a crucial task for researchers to effectively remove heavy metals from the different wastewater, and reuse heavy metal energy sources. Advances in this type of work are well underway. The aim of this class of research is not only to prevent the contamination of heavy metals in freshwater bodies, but also to use them in different applications after disposal. Among them, it has been found that removed heavy metals used in the field of catalysis for the synthesis of different chemicals are attractive. Much research has already been published and, among them, several interesting works have been discussed on in this review. Godiya et al., fabricated a porous three-dimensional (3-D) sodium alginate (ALG)/polyethyleneimine (PEI)-based composite hydrogel for the absorption of Cu(II) and Pb(II) from wastewater. Later, the in situ fabricated hydrogel reduced metal ions into metal nanoparticles, and metal nanoparticle-based hydrogel systems were used in the catalytic reduction of toxic 4-nitro phenol (4-NP) in presence of sodium borohydride (NaBH_4_) as a reducing agent [175]. Ma et al., used fly ash and sawdust for the absorption of the aqueous solution of Ni(II) ions, and the Ni(II) ion-absorbed system was used for catalytic degradation of 2-chlorophenol (2-CP) in an ozone oxidation batch reactor [176]. Giri et al., described the electron dense polypyrrole-mercaptoacetic acid (PPy-MAA) composite for the absorption of Ag(I) from the silver-contaminated aqueous solutions. Subsequently, the silver ion was reduced in situ by the reducing agent in the composites, and the nanoparticle-decorated composites were used for the catalytic reduction of various toxic nitro aromatic compounds. The results show that the composite can effectively catalytically reduce nitro aromatic compounds, and they can be recycled up to tens of cycles without much loss in activity [177]. Godiya et al., prepared an inexpensive and sustainable silk fibroin (SF)/polyethyleneimine (PEI) composite hydrogel, and employed it in the absorption of heavy metals such as Cu(II), Pb(II), Cd(II), Zn(II), Ni(II), and Ag(I) ions from aqueous solution. Later, Cu(II) and Ag(I) ion-absorbed hydrogel was reduced in-situ by NaBH_4_ as a reducing agent to form metallic copper and silver nanoparticles, and the fabricated nanoparticle-decorated composites exhibited excellent antimicrobial as well as catalytic activity for the formation of 4-aminiphenol (4-AP) from 4-NP [178]. Das et al., synthesized polypyrrole with mercapto-functionalized chelating groups (PPy/MAA) through a novel approach for the absorption of silver ion from aqueous solution and, subsequently, the silver ion was reduced to metallic silver (Ag) to fabricated PPy/MAA/Ag^0^ composites. After that, the prepared nanoparticle-decorated composites were successfully applied in antimicrobial activity, catalytic reduction of 4-NP, and NO_2_ gas detection. The synthesized PPy/MAA/Ag^0^ composites exhibited excellent activity in all applications mentioned above [179]. Das et al., in prepared thiol-functionalized polypyrrole (PPy/MAA) composite, and used it for the absorption of highly toxic Hg(II) from the waste industrial effluent. Fabricated composites with heavy metal ions absorbed were applied in the catalytic hydrogenation of phenylacetylene. The catalytic reduction reaction was performed in presence of a 5 mol% fabricated catalyst at 90 °C for 6 h, and the catalyst converted phenylacetylene into acetophenone with 55% yield [37]. In another work, Sun et al., self-assembled homopolymer vesicle from poly(amic acid) (PAA) and the vesicle employed in the absorption of polycyclic aromatic hydrocarbons (PAHs), cationic dyes, and heavy metal ions (such as Ni(II) and Ag(I) ions) of the wastewater solution. The self-assembled vesicle absorbed the three pollutants mentioned above through the π-π interaction, the hydrophobic effect, and the electrostatic interaction. Later, the self-assembled absorbed of Ag(I) was reduced in situ in metallic silver nanoparticles in the presence of NaBH_4_ as a reducing agent, and the illuminated nanoparticles of self-assembled PAA vesicle exhibited excellent catalytic activity towards the reduction of 4-NP, and can be recycled ten times without significant loss in activity [180]. Javed et al., synthesized three types of hydrogel; one is anionic [Poly(methacrylic acid) (P(MAA))], another is neutral [poly(acrylamide) (P(AAm))], and the last is cationic hydrogels [poly(3-acrylamidopropyltrimethyl ammonium chloride) (P(APTMACl))]. The hydrogels were used for the absorption of heavy metal such as Cu(II), Co(II), Ni(II), and Zn(II) ions from aqueous solutions, and found the absorption of the following order Cu(II) > Ni(II) > Co(II) > Zn(II). The hydrogels absorbed with heavy metal ions using the Freundlich and Langmuir isotherm absorption mechanism. After that, the heavy metal ions absorbed on the hydrogel were transformed into metal nanoparticles, and the nanoparticle-formed hydrogel systems were used for catalytic hydrogenation of dyes such as methylene blue, methyl orange (MO), Congo red (CR), and nitro aromatics such as nitrobenzene and 4-nitrophenol [181]. Su et al., prepared an effective geopolymer microsphere (Na-SGS) using NaOH activated slag, and applied it in the absorption of Ni(II) ions from an aqueous solution of Ni(II) salts. They extensively studied the absorption properties of Ni(II) ions by the microsphere as a function of pH, absorbent dose, temperature, contact time, and initial concentration of Ni(II) salts in aqueous solutions. Later, the absorbed systems were reduced directly to convert it to supported Ni catalyst, and employed in the catalytic activity for the methanization of carbon dioxide. The supported catalyst exhibits the highest catalytic conversion at 300 °C, and the performance of the catalyst was better at a lower temperature, around 100 °C. The process of synthesis active absorbent materials and the subsequent use of materials as a catalyst is schematically represented in Figure 4. The methods used for the preparation of highly active absorbents and subsequently reduced materials used for catalytic conversion can be one of the valuable and recyclable metal resources to protect our environment [182,183]. The strategy of microspheres when dealing particularly with microfluidically generated chitosan microspheres came not only to the fixation of heavy metals, but also to their selectivity [184]. Wang et al., synthesized sulfur atom-doped porous biomass through the one pot pyrolysis method, and the prepared biomass shows excellent absorption capability towards Cu(II) and Ni(II) ions from mixed heavy metal containing aqueous solutions. This metal ion immobilized porous biomass became a catalyst through an in situ reduction process. The manufactured catalyst demonstrated excellent catalytic activity towards the degradation of methylene blue (MB) as model dye pollutants and the reduction of toxic Cr(VI) ions. This might be due to the formation of active SO_4_^−·^ ions radicals that promote the degradation reaction, whereas the formation of active hydrogen radicals accelerates the reduction reactions [185]. Kakaei et al., prepared triazole and triazolium ligands modified clinochlore clays, and used them for the study of absorption efficacy of heavy metals such as Pb(II), Co(II), and Zn(II) ions from the mixture of the corresponding salts. They also studied the effect of pH on the kinetics of the absorption. Finally, to study the catalytic efficacy towards reductions in aromatic nitro derivatives, the absorbed modified clays of Co(II) ions were reduced to make a nanocatalyst [186].

Liu et al., used a calcined product of Mg/Al-CO_3_ hydrotalcite (referred to as LDO) for the absorption of Cu(II) and Cr(VI) ions from the mixed solution, and later it was calcined at 500 °C for four hours to fabricate a Cu- and Cr-decorated LDO catalyst. The efficiency of the fabricated catalyst was determined by the degradation of methyl orange (MO) in the presence of hydrogen peroxide. They reported that the presence of Cu and Cr as an element enhanced the degradation efficiency of the fabricated catalyst [187]. Ivanets et al., applied magnesium ferrite (MgFe_2_O_4_) in the absorption of four different heavy metals ions such as Mn(II), Co(II), Ni(II), and Cu(II) from their corresponding aqueous salts solutions. The various metal ion-loaded MgFe_2_O_4_ were used as a Fenton-like catalyst towards MB degradation in the presence of hydrogen peroxide. They used two different concentrations of metallic salts for the absorption MgFe_2_O_4_, and found a different order for various concentrations. At high concentrations, absorbed metal ions MgFe_2_O_4_ show another trend, as compared to low concentrations. The results of catalytic activity at high concentration follow the order of MgFe-Mn(II) > MgFe-Ni(II) > MgFe-Co(II) > MgFe-Cu(II), whereas at low concentration, the results follow the order of MgFe-Cu(II) > MgFe-Mn(II) > MgFe-Co(II) > MgFe-Ni(II). They revealed that the variation in results in the catalytic activity towards MB degradation might be due to increased mobility of Fe(III) and metal ions and the catalyst can be reused up to four cycles without significant loss in its activity [188]. The use of the recycling of valuable materials recovered from wastewater and involved in catalysis reactions has been promoted with the aim of improving the recovery rate and reducing environmental pollution caused by these metals. In addition, applications of these recovered materials can solve the problems of resource use in the field of catalysis for industrial purpose.

## 8. Research Challenges

There are many conventional methods available to treat these pollutants, as mentioned earlier; however, the techniques are more expensive. Among them, it has been found that the absorption method is a cheap and easy process to eliminate heavy metals from wastewater solutions [189]. Significant progress must still be made on heavy metals, but research based on the use of heavy metals after disposal has been in the early stages, and the growth of this research is very slow. Significant progress has still been made in heavy metal work, but research based on use of heavy metals after disposal has been the initial stage, and the growth of this research is also very slow. On the other hand, heavy metals released from various wastewater sources are converted into toxic nanomaterials, either by abiotic or biotic interaction, which has a strong impact on environmental pollution, as the converted nanoparticles have a high absorption capacity of pollutants due to their high specific surface area. Therefore, it is critical to treat these heavy metals before their transformation, or to design a method by which heavy metals and nanoparticles (NPs) can be removed simultaneously [190]. Another challenge is that the materials used for the recovery of heavy metals are not soluble in aqueous systems, and, according to the theory of mass transfer, in a two-phase system, heavy metals must come into contact with the removal system to discard the pollutants from wastewater. This problem can be overcome through the functionalization of the removal system to make it hydrophilic in nature and, in the future, research should focus more on this problem [191]. It has been found that in wastewater there are many essential metals competing with heavy metals. Therefore, during the treatment of heavy metals from wastewater obtained from various sources, it should be considered that the essential elements should not be removed together with the heavy metals. It has been a great task to design the system by which we can specifically discard toxic heavy metals from wastewater without discarding essential elements [192]. In addition, one thing to allow for during the removal process is that although heavy metals are toxic in nature, several heavy metals are required for life (which has already been discussed). Therefore, heavy metals discarding systems should be designed in such a way so that the systems remove heavy elements to a particular concentration not below or above this critical concentration [193]. Therefore, in brief these aforementioned challenges must be urgently overcome to reduce the environmental pollution as well as the use of heavy metals for different fields of applications [194].

## 9. Conclusions

Heavy metals are precious, and are widely used in various catalysis fields of industry. Due to the insignificant natural sources and the increase in the field of application, as well as the economic cost, it is critical to recycle valuable materials as a secondary resource. In the last decade, the amount of recycling of heavy metals recovered from wastewater is very limited. As time goes on, research on the recovery of valuable heavy metals from wastewater and used as secondary energy resources specifically in the field of catalysis has increased. Although technologies for the recovery of precious metals have been developed over time, there are still some challenges that need to be overcome in the future. In this review article, we have discussed the sources of heavy metals that pollute the environment, and the negative impacts of heavy metals on human health in terms of the optimal concentration of heavy metals in freshwater bodies. In addition, the rules and regulations of the optimal level of heavy metal concentrations provided by the different known organizations are briefly summarized. After that, we demonstrated the conventional methods as well as the recently developed technologies for the treatment of toxic heavy metals present in the wastewater. In addition, there is also discussion on the quality of the freshwater bodies when it is affected by toxic heavy metals and their correlation with various parameters such as pH, temperature, ionic strength, and natural organic matter. Finally, the article concludes with the reuse of heavy metals by various systems, and uses precious metals in different fields of catalytic reactions. In addition, we focus on the challenges that researchers face during the recovery and reuse of heavy metals in different fields of applications. Therefore, future research in this field should also scrutinize all the effects of different water conditions on the removal of heavy metals, and determine the activity of these precious heavy metals in a broader perspective from the field of catalysis research.

## Data Availability

No new data were created or analyzed in this study. Data sharing is not applicable to this article.

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
