# Peer review of "Review on the Use of Heavy Metal Deposits from Water Treatment Waste towards Catalytic Chemical Syntheses"

_ijms, 2021, doi:10.3390/ijms222413383_

Round 1
Reviewer 1 Report
1. Rephrase the sentence " The trend in recent research trend" in abstract
2. " In 1987, Environmental Protection Agency (EPA)" I suggest authors to look for new regulation of EPA and revise the information.
3. Remove the etc form the four cateegories, if there is some thing extra information this review article should have that information.
4. " According to various survey reports, developing countries are mostly affected by heavy
metal pollution [16]", since authors claiming based on various reports, it should have replicated in the references too.
5. "e heavy metals found in the mining areas are Fe, Mn, Cu, Zn, Pd, Co, As, Ni and Cd", is it particular on specific mining dumpsite or overall statement? clarify
6. The description about electronic waste is not much informative relative to the content. authors can include some recent informations on electronic waste.
7. Under the power plants only coal was discribed, what abou the other powerplants like nuclear?
8. In my opinion industrial and mining waste can be combined and can be correlated.
9.Also, any schematic digram potrays the information of different souurces would be more suitable.
10. If authors tries to include the overall statement of hevaymetal concentration in wastewater, they should have tried some systematic approach, not as followed by the other information. for eg, the statement " The order of concentration of heavy metals in the wastewater is as follows:
Fe>Zn>Cu>Ni>Mn>Pb>Cr>Cd [50]", this information is based on the data from 2003 and there might be rigorous change in the environment as well as in the order of concentartion. it should be carefully noticed.
11.Fig 1 can be modified little without any confusion on lines.
12. Authors could have tabulated the advantages and disadvantages of treatment process, so the readers can be more informative when comparing the methods.
13.Research challenges are not critically analysed from the literature.
Author Response
Please, see the attachment.

Reviewer 2 Report
Manuscript ID: ijms-1470492
Title: Review on Use of Heavy Metal Deposits from Water Treatment Waste towards Catalytic Chemical Syntheses
Authors: Albert Poater et al.
Section 2.4. Where are more heavy metals concentrated in coal fly ash (CFA) or coal bottom ash (CBA)? Can the authors provide these data? Since the waste of coal-fired power plants is divided approximately like this: 90% CFA, 10% - CBA.
In section 2. Authors write: (6) industrial sources, (7) household waste, and (8) other resources. However, section 2.7 wrote – Domestic waste. Is it correct information?
Section 2. Why did the authors not indicate the main pollutants - Metallurgy refineries that produce heavy metals and gold (in the case of arsenic, mercury, and zinc)? Nature is polluted both by the refineries themselves and by the landfills, sludge stores, etc. from these refineries. The authors should introduce a new section that will describe the refineries of ferrous and non-ferrous metallurgy.
Section 3.1 and Section 3.2 Authors write about heavy metals concertation in sludge and wastewater, however, didn’t write any numbers of wt%, mg/kg, ppm, g/L, mg/L, etc. Why?
Section 5. This section describes different methods for heavy metals removal from wastewater. However, the authors didn’t write any numbers about the efficiency of removal (in %) for these methods. Why?
Section 5. What’s about different types of zeolites application for sorption metals from water?
Section 6. Add more numbers of different parameters and their effect on removal efficiency.
The conclusions are too general, make a few points: 1) 2) 3), etc.
In almost all sections of the review, the authors do not use numerical values. It is necessary to add to the article more data on the removal efficiency of heavy metals, the concentration of metals in various phases, data on the technological parameters (T, the concentration of reagents, process time, liquid to solid ratio, etc.) of the metal’s removal processes, etc. Add more figures and tables.
Author Response
Please, see the attachment.

Reviewer 3 Report
The manuscript related to Review on Use of Heavy Metal Deposits from Water Treatment Waste towards Catalytic Chemical Syntheses is an interesting. But manuscript need major revision.
- Provide the statement describing one or more key hypotheses that the work described in the manuscript was intended to confirm or refute. Inclusion of a hypothesis statement makes it simple to contrast the hypothesis with the most relevant previous literature and point out what the authors feel is distinct about the current hypothesis (novelty).
- Under section 5.3. Adsorption Process author have mention “Adsorption is also a well-known method used to reduce the concentration of metals in wastewater”. Provide with the relevant references to support the above sentence. [International Journal of Biological Macromolecules 143, 60-75, 2020; Carbohydrate Polymers 134, 646–656, 2015]
- Kindly remove section 5.3. a Activated Carbon (AC), 5.3. b Carbon Nanotubes (CNTs), 5.3. c Wood sawdust. Because enough of information on the above topic are available in literatures.
- Under section 5.4. Current Methods, author have mention “ Many current technologies have been developed, and among them membrane filtration, photocatalysis and nanotechnology, the most popular methods for removing heavy metals from wastewater”. I do not feel it is appropriate to say nanotechnology as a current methods.
- Sectiion 5.4. c Nanotechnology , author have mention “The materials that have a dimension in the range of nanometer exhibit distinguished properties compared to that of their corresponding bulk counterpart. Nanomaterials due to their high specific surface area are extensively applied in the treatment of heavy metals present in wastewater”. But not a single references are added in it support. Kindly add few reference. [Cancers 13 (9), 2214, 2021]
- The main and fundamental purpose of writing a review is to create a readable synthesis of the best resources available in the literature for an important research question or a current area of research. Although the idea of writing a review is attractive, it is important to spend time identifying the important questions. In a systematic review with a focused question, the research methods must be clearly described.
- Discuss more on the advance oxidation process for the wastewater treatment and also provide with the mechanism of action.
- Section 6.1. Effect on pH, required more in depth discussion.
- Discuss more advance nanomaterials used for wastewater treatments such as hydrogel nanocomposite, carbon quantum dots, graphene composite etc.
- Harmful impact of metal ion to the human being need more in depth discussion.
- What are the major drawback associate with wastewater need to be added.
- Most of the section in this script is not study in depth. I will suggest the author to discuss more details each section incorporated in this review.
- Separate section for plausible mechanism for the action reported in the literature need to be added and discuss in the text.
- Please write your text in good English (American or British usage is accepted, but not a mixture of these). English language manuscript required editing to eliminate possible grammatical or spelling errors and to conform to correct scientific English.
Author Response
Please, see the attachment.

Reviewer 4 Report
The manuscript entitled “Review on Use of Heavy Metal Deposits from Water Treatment Waste towards Catalytic Chemical Syntheses” aims to review the types of heavy metals obtained from wastewater and their recovery through commonly practiced physico-chemical pathways. This topic is well known but important. Still, this manuscript is full of linguistic and material errors. In my opinion, it should be carefully rewritten in order to be published.
Some of my concerns:
Non-degradability of heavy metals is the wrong term. Maybe use persistence instead?
The introduction is so confusing, and the sentences are too long. It has to be improved. Some examples are copied below.
“The discharge of effluents from industrial waste, specifically heavy metals containing waste, into freshwater bodies has gained more attention due to the toxic effect on the life cycle of living beings [2].”
“Other types of pollutants such as plastics, plant nutrients, pathogens, and synthetic organic & inorganic chemicals are also present in wastewater and these pollutants are not so much harmful to the environment as heavy metals [3].”
Mistakes like naming the Mg and Na heavy metals are intolerable. The classification of heavy metals should be reconsidered with the utmost attention. Also, heavy metals are those with
- Effect of water quality on heavy metals and their removal – it is not about quality, but properties.
6.1. Effect on pH… Do you mean Effects of?
“The recovery and reuse of heavy metals from wastewater is the main obstacle that must be overcome to free our environment from pollution.” It is hardly the main obstacle. Do not use such strong statements, especially if they are not true.
Discussion Ag as toxic metal is unnecessary. Instead, it would help to focus more on the harmful effects of heavy metals on the nervous system. An in-depth discussion is generally missing.
Author Response
Please, see the attachment.

Reviewer 5 Report
The paper presents the review on the use of heavy metal deposits from water treatment waste towards catalytic chemical syntheses. The presentation of methods and scientific results in the current form is satisfactory for publication in the IJMS journal. The minor drawbacks to be addressed can be specified as follows:
1. Tab. 1. (i) Please sort heavy metal alphabetically, unless; there is another reason for their order. (ii) Name of organization ---> Name of Organization.
2. “4. Harmful effect of heavy metals on human health”. Unfortunately, the authors did not mention manganese in this section.
3. Fig. 1. (i) Carbon Nanotubes ---> Carbon Nanotube. (ii) Adsorption is also physico-chemical process!!!
4. “5.2. Physico-chemical Process” ---> “5.2. Physico-Chemical Process” see Fig. 1. See also “5.2. b Ion exchange” and “5.3. c Wood sawdust”. Capital letters at the beginning words similarly as in Fig. 1!!!
5. Page 9, Eq. (3). (i) Dots after H+ in the reaction equations? This form of writing “(Resin) (Metal ions)(Resin) (Solution)” is not entirely understandable and clear.
6. 5.3. Adsorption Process. What about other forms of carbon, e.g. graphene?
7. “The specific surface area of AC varies from 1260 to 3250 m2 gm-1”. Where does this range of values come from? Typical ACs values are less than 1,200 m2/g (Norits) and are used for water treatment and metal removal, including biological films.
8. What about the biological films and their use in removing metals - for example, manganese?
9. 6.3. Effect on ionic strength. Capital letters at the beginning words.
10. Graphics – file: ijms-1470492-non-published.pdf. Please change the font colour in the word “Separator”, i.e. yellow ---> black.
Author Response
Please, see the attachment.

Reviewer 6 Report
Authors presented review, the source, toxicity limits, treatment methods and reuse techniques of heavy metals polluting wastewater are compiled in an orderly manner, with an understanding of the properties of heavy metals, which illustrate the status current treatment methods and future prospects in this research area. This presentation of the issue is very interesting and shows the possibility of reusing metal waste. The authors prepared the review publication on the basis of a number of publications, uncritically accepting the research results presented there. This approach causes many inaccuracies in the paper.
- The aluminum with a density of 2.7 g/cm3 is not a heavy metal, despite the fact that it is copied in many lists there.
- The metals in terrestrial conditions, apart from nuclear reactors, are not transformed into other elements; therefore they are never biodegradable under these conditions. Since the paper discusses the total concentration of metals, not their speciation forms, the term biodegradable is incorrect. Chemical compounds, not metal atoms, can be biodegradable.
- The phrase "Heavy metals are highly soluble in aqueous media ....." are incorrect heavy metals not dissolve in aqueous media, dissolve only their compounds. Throughout the publication, it should be specified what the authors understand by the concept of materials. Metals, or their compounds or sum of individual metals and metal compounds is discussed in paper.
- For the publication to be substantively correct, the entire text should be corrected in terms of nomenclature. What the authors in the original papers studied metals, metal compounds or discussed the sum of all forms of metals.
In this type of review in 2021, the sum of metal compounds in the environment should no longer be discussed, but discussed about specific speciation forms and equilibria between them.
One should stop perpetuating the belief in readers that metals such as copper, iron ... are toxic, while their compounds are primarily toxic.
He suggests that the authors of the publication make changes in the review paper from this angle.
Author Response
Please, see the attachment.

Round 2
Reviewer 2 Report
The authors answered all questions in detail and made major revision to the article.
In this form, article "Review on Use of Heavy Metal Deposits from Water Treatment Waste towards Catalytic Chemical Syntheses" can be accepted.
Reviewer 3 Report
The authors have satisfactory revised the manuscript.
Reviewer 4 Report
I recommend this manuscript for publication in present form .